# Learning Lyapunov-Stable Polynomial Dynamical Systems Through Imitation

**Amin Abyaneh**
Department of Electrical and Computer Engineering
McGill University
`amin.abyaneh@mail.mcgill.ca`

**Hsiu-Chin Lin**
School of Computer Science
McGill University
`hsiu-chin.lin@cs.mcgill.ca`

**Abstract:** Imitation learning is a paradigm to address complex motion planning problems by learning a policy to imitate an expert's behavior. However, relying solely on the expert's data might lead to unsafe actions when the robot deviates from the demonstrated trajectories. Stability guarantees have previously been provided utilizing nonlinear dynamical systems, acting as high-level motion planners, in conjunction with the Lyapunov stability theorem. Yet, these methods are prone to inaccurate policies, high computational cost, sample inefficiency, or quasi stability when replicating complex and highly nonlinear trajectories. To mitigate this problem, we present an approach for learning a globally stable nonlinear dynamical system as a motion planning policy. We model the nonlinear dynamical system as a parametric polynomial and learn the polynomial's coefficients jointly with a Lyapunov candidate. To showcase its success, we compare our method against the state of the art in simulation and conduct real-world experiments with the Kinova Gen3 Lite manipulator arm. Our experiments demonstrate the sample efficiency and reproduction accuracy of our method for various expert trajectories, while remaining stable in the face of perturbations.

**Keywords:** Imitation learning, Safe learning, Motion planning, Dynamical system, Semidefinite programming, Robotic manipulation

## 1 Introduction

Motion planning for robotic systems is generally regarded as a decomposition of a desired motion into a series of configurations that potentially satisfy a set of constraints [1]. Imitation learning tackles motion planning by imitating an expert's behavior to learn a planning policy [2]. To this day, only a handful of imitation learning methods provide mathematical stability guarantees for their resultant policy. Stability is a critical factor when deploying imitation policies in environments exposed to external perturbations. Therefore, unpredictable environments require a policy that reasonably responds in unexplored regions of state space, away from original demonstrations.

Researchers have turned to autonomous dynamical systems (DS) as a means to learn stable motion planning policies [3, 4, 5]. Essentially, a parametric time-invariant DS is optimized to provide an action (velocity) given the current state (position), while adhering to constraints that attain global Lyapunov stability. This approach leads to safety and predictability of planned trajectories, even in areas of state space without expert demonstrations. However, previous work is mostly confined to basic Lyapunov functions that adversely impact the reproduction accuracy, and require sufficiently large set of demonstrations. Others have proposed approaches focused on diffeomorphism and Riemannian geometry [6, 7, 8] and contraction theory [9], that are prone to quasi-stability, increased computational time, or restricted hypothesis class.

We propose a method to simultaneously learn a polynomial dynamical system (PLYDS) and a polynomial Lyapunov candidate to generate globally stable imitation policies. Polynomials, depending on

7th Conference on Robot Learning (CoRL 2023), Atlanta, USA.

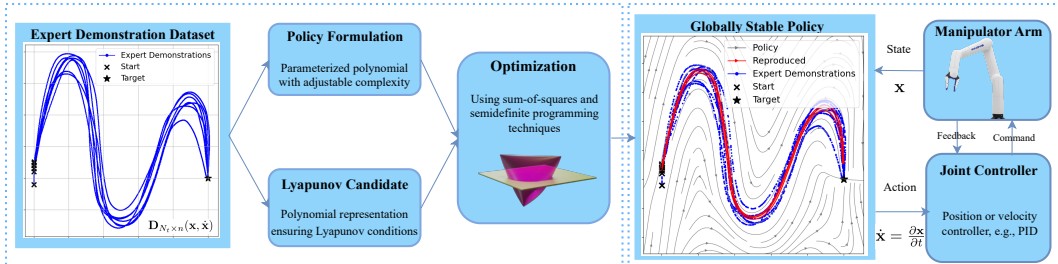

Figure 1: Overview of the stable policy learning framework. Policy learning (left) optimizes a stable polynomial DS from expert's demonstration data. This policy is then deployed (right) to plan globally stable and predictable trajectories in the entire state space.

the degree, possess an expressive power to approximate highly nonlinear systems, and polynomial regression can empirically compete with neural networks on challenging tasks [10, 11]. Unlike most neural policies, global stability can be naturally expressed with polynomials. Polynomials also enable us to utilize efficient semi-definite programming [12, 13] and sum-of-squares (SOS) optimization techniques [14, 15], and offer adaptability to expert's demonstrations.

Our main contribution is twofold. We propose a polynomial representation of the planning policy and Lyapunov candidate function, coupled with concrete mathematical stability certification for precise and safe replication of the expert's demonstrations, as depicted in Figure 1. Then, we define a regularized semi-definite optimization problem to jointly learn the DS and the Lyapunov candidate with higher flexibility and precision. We compare the reproduction accuracy of PLYDS with alternatives in the literature and evaluate the performance in both simulation and real robotic systems.

## 2  Background and Notation

Consider a system operating in a state-space $\mathcal{X} \subset \mathbb{R}^n$, e.g., a robot in its task- or configuration-space. The system can execute actions in $\mathcal{A} \subset \mathbb{R}^n$, for instance, velocity or torque commands, leading to state evolution. We denote the state variable with $\mathbf{x} \triangleq [x_1 \ x_2 \ \ldots \ x_n]^T \in \mathcal{X}$, and consider the action variable to be the state's derivative $\dot{\mathbf{x}} \in \mathcal{A}$. Within this space, our goal is to learn an imitation policy through a dataset of experts' state-action pairs, referred to as trajectories.

Let $N_d \in \mathbb{N}$ be the number of trajectories demonstrated by the expert. Each trajectory contains $N_s \in \mathbb{N}$ state-action pairs. The dataset of expert trajectories stacks all state-action pairs, defined as:

$$\mathbf{D} \triangleq \Big\{ \big(\mathbf{x}^d(s), \ \dot{\mathbf{x}}^d(s)\big) \ \big| \ d \in \{1, \ldots, N_d\}, \ s \in \{1, \ldots, N_s\} \Big\}, \tag{1}$$

where $(\mathbf{x}^d(s), \ \dot{\mathbf{x}}^d(s))$ is the dataset entry corresponding to the $s$-th sample of the $d$-th demonstrated trajectory. The dataset $\mathbf{D}$ holds $N_t = N_d N_s$ samples. We assume that the trajectories contain the same sample size ($N_s$), share a common target ($\mathbf{x}^* \in \mathcal{X}$), and have zero velocity at the target, i.e., $\mathbf{x}^d(N_s) = \mathbf{x}^*$ and $\dot{\mathbf{x}}^d(N_s) = \mathbf{0}$ for all trajectories $d \in \{1, \ldots, N_d\}$.

**Definition 2.1.** *(Dynamical Systems). The mapping between the state and the action in each sample can be modelled with a time-invariant autonomous dynamical system (DS), denoted by:*

$$\dot{\mathbf{x}} = f(\mathbf{x}) + \epsilon = \hat{f}(\mathbf{x}), \quad f, \hat{f} : \mathcal{X} \to \mathcal{A}. \tag{2}$$

In Equation (2), $f$ is an ordinary differential equation for the true underlying DS. The term $\epsilon \in \mathbb{R}^n$ captures measurement and recording noise of expert's demonstrations. We assume that $\epsilon$ is embedded in the estimated DS, $\hat{f}$, and eliminate the need for modeling its distribution. Following [3], we aim at learning a noise-free estimation of $f(\mathbf{x})$, denoted by $\hat{f}(\mathbf{x})$. One can view $\hat{f}(\mathbf{x})$ in Equation (2) as a *policy* that maps states to actions for reproducing the demonstrated trajectories in the state-space. For instance, when the robot is located in $\mathbf{x}_0 \in \mathcal{X}$, the policy yields an action $\dot{\mathbf{x}}_0 = \hat{f}(\mathbf{x}_0)$, which can be passed to the robot's velocity controller.

The estimated DS in Equation (2), $\hat{f}(\mathbf{x})$, is globally asymptotically stable (GAS) around an equilibrium point $\mathbf{x}^e$, if and only if for every initial state, $\mathbf{x} \to \mathbf{x}^e$ as the system evolves and time goes to infinity [16]. A popular tool to study the GAS property of a DS is the Lyapunov stability theorem. According to this theorem, a DS exhibits GAS if there exists a positive-definite function $v : \mathcal{X} \to \mathbb{R}$, known as Lyapunov potential function (LPF), such that $\dot{v}(\mathbf{x}) < 0$ for all $\mathbf{x} \neq \mathbf{x}^e$ and $\dot{v}(\mathbf{x}^e) = 0$. To ensure GAS for $\hat{f}(\mathbf{x})$, we simultaneously learn the policy, $\hat{f}$, and the LPF, $v$.

## 3 Related work

Extensive research is conducted on imitation learning and its applications in robotic motion planning for a variety of tasks. Existing efforts can be divided into the following predominant research tracks.

**Dynamical systems for motion planning.** Dynamical systems have proved to effectively counter autonomous motion planning problems by proposing a time-invariant policy [17]. Traditional methods of encoding trajectories are based on spline decomposition [18], Gaussian process regression [19], or unstable dynamical systems [20, 21]. They either lack robustness because of time-variance or fail to provide GAS. SEDS [3] is the first attempt to learn stable planning policies. However, its performance declines when applied to highly nonlinear expert trajectories. Most notably, it suffers from trajectories where the distance to the target is not monotonically decreasing. The intrinsic limitation of SEDS comes from the choice of a simple Lyapunov function. Follow-up research introduces more complex Lyapunov candidates to stably mimic nonlinear trajectories [4, 22], but are still restricted in representing the Lyapunov candidate. Others have tried to tackle SEDS limitations through diffeomorphic transformations and Riemannian geometry [6, 8, 7] that yield quasi-stable planners for some trajectories, and contraction theory [9] that restricts the class of metrics to make the optimization tractable. Lastly, most improvements to the original SEDS still use the Gaussian mixture model formulation, that is vulnerable in presence of limited expert demonstrations.

**Imitation learning.** Recent imitation learning developments can be applied to motion planning tasks with minimal modifications, since motion planning can be achieved by finding a (not necessarily stable) policy in the robot's task-space from the expert's behavior. For instance, GAIL [23] introduces an adversarial imitation learning approach that directly optimizes the expert's policy, but requires a large set of expert's data (low sample efficiency) and extensive training iterations. The growing interest in neural policies has also led to the development of end-to-end autonomous driving [24] and behavioral cloning [25, 26, 27] methods. Nevertheless, they generally lack GAS, and it is unclear whether the robot can recover from perturbations. The same drawbacks exist with apprenticeship learning approaches, such as Abbeel and Ng [28] and inverse reinforcement learning, such as Ziebart et al. [29], and the computational demand is even higher for the latter.

**Stability in neural dynamical systems.** Methods such as [30, 31] represent the dynamics with a Neural Network, and propose the joint training of dynamics and a Lyapunov function to guarantee the stability. Though theoretically sound, these methods have only been applied to rather simple settings and require large demonstration sets. Neural Lyapunov methods [32, 33, 34] promise a data driven and potentially stable approach to control and model nonlinear dynamics, but lack global stability. Methods such as [35] are also not stable-by-design and the dynamical system lacks autonomy.

## 4 Methodology

We instantiate the policy and the corresponding LPF candidate, $\hat{f}$ and $v$, with two polynomials in Section 4.1 and Section 4.2, respectively. This allows us to accurately imitate an expert's behavior, while providing formal GAS guarantees. Subsequently, we formulate a tractable optimization problem for jointly learning the policy and the LPF in Section 4.3.

### 4.1 Dynamical system policy formulation

We need to approximate the unknown underlying DS in Equation (2) to discover the mapping between states and actions from expert's behavior. To this end, we opt to model the policy with a

parametric polynomial. The representative power of polynomials was originally established through the Weierstrass approximation theorem, stating that every continuous function defined on a closed interval can be approximated with desired precision by a polynomial. This idea is fortified by recent studies, such as [10, 11], that compare polynomials to neural networks on a variety of tasks.

**Definition 4.1.** *(Polynomial Dynamical Systems). A Polynomial Dynamical System (PLYDS) is a polynomial approximation of the policy in Equation* (2)*, and is expressed as,*

$$\dot{\mathbf{x}} = \hat{f}(\mathbf{x}; \ \mathbf{P}) \triangleq \begin{bmatrix} \mathbf{b}_{\mathbf{x},\alpha}^T \mathbf{P}_1 \mathbf{b}_{\mathbf{x},\alpha} & \mathbf{b}_{\mathbf{x},\alpha}^T \mathbf{P}_2 \mathbf{b}_{\mathbf{x},\alpha} & \dots & \mathbf{b}_{\mathbf{x},\alpha}^T \mathbf{P}_n \mathbf{b}_{\mathbf{x},\alpha} \end{bmatrix}^T, \tag{3}$$

*where* $\mathbf{b}_{\mathbf{x},\alpha} \triangleq \begin{bmatrix} 1 & (\mathbf{x}^T)^{\circ 1} & (\mathbf{x}^T)^{\circ 2} \dots (\mathbf{x}^T)^{\circ \alpha} \end{bmatrix}^T$ *is the polynomial basis of degree* $\alpha \in \mathbb{N}$*, and* $(\mathbf{x}^T)^{\circ k}$ *is the element-wise* $k$*-th power of* $\mathbf{x}^T$*. Every row* $i$ *of* $\hat{f}$ *is a polynomial of degree* $2\alpha$*,* $\hat{f}_i(\mathbf{x}; \ \mathbf{P}_i) = \mathbf{b}_{\mathbf{x},\alpha}^T \mathbf{P}_i \mathbf{b}_{\mathbf{x},\alpha}$*, where* $\mathbf{P}_i \in \mathbb{S}^{\alpha n+1}$ *and* $\mathbb{S}^k \triangleq \{S \in \mathbb{R}^{k \times k} | S^T = S\}$*. The matrix* $\mathbf{P} \in \mathbb{S}^{\alpha n^2 + n}$ *encapsulates the block-diagonal form of all* $\mathbf{P}_i$ *matrices.*

Below, we present an example to show how PLYDS, as defined in Definition 4.1, captures nonlinear time-invariant policies. One can further complicate the policy by increasing $\alpha$, which in turn produces a larger basis vector and a more flexible polynomial.

**Example 4.1.1.** *A second-order polynomial representation of a one-dimensional DS is:*

$$\dot{\mathbf{x}} = \hat{f}(\mathbf{x}; \ \mathbf{P}) = \begin{bmatrix} \mathbf{b}_{\mathbf{x},\alpha}^T \ \mathbf{P}_1 \ \mathbf{b}_{\mathbf{x},\alpha} \end{bmatrix} = \begin{bmatrix} 1 & x \end{bmatrix} \begin{bmatrix} p_{00} & p_{01} \\ p_{01} & p_{11} \end{bmatrix} \begin{bmatrix} 1 \\ x \end{bmatrix} = p_{00}x^2 + (p_{01} + p_{10})x + p_{00},$$

*where* $\alpha = 1, \mathbf{b}_{\mathbf{x},\alpha} = [1 \ x]^T$*. Note how* $\mathbf{P}$ *can be symmetric without loss of generality.*

## 4.2 Global stability guarantees for polynomial dynamical systems

As explained in Section 4.1, a polynomial policy allows for accurately imitating the expert's demonstrations. Yet, there is no formal GAS guarantee that the robot will ultimately converge to the target in the face of perturbations, deflecting it from the expert's trajectories. Owing to the Lyapunov stability theorem, finding an LPF that meets the criteria in Section 2 ensures the desired stability [36].

The major challenge lies in learning an LPF, $v$, which is a positive definite function with negative gradient. We tackle this by confining to the class of polynomial LPF candidates.

**Definition 4.2.** *(Polynomial Lyapunov Candidate). A multidimensional polynomial LPF is given by,*

$$v(\mathbf{x}; \ \mathbf{Q}) \triangleq \begin{bmatrix} \mathbf{b}_{\mathbf{x},\beta}^T \mathbf{Q}_1 \mathbf{b}_{\mathbf{x},\beta} & \mathbf{b}_{\mathbf{x},\beta}^T \mathbf{Q}_2 \mathbf{b}_{\mathbf{x},\beta} & \dots & \mathbf{b}_{\mathbf{x},\beta}^T \mathbf{Q}_n \mathbf{b}_{\mathbf{x},\beta} \end{bmatrix}^T, \ v : \mathcal{X} \to \mathbb{R}^n, \tag{4}$$

*where* $\beta \in \mathbb{N}$ *is the polynomial basis degree. Each row is defined by* $v_i(\mathbf{x}; \ \mathbf{Q}_i) = \mathbf{b}_{\mathbf{x},\beta}^T \mathbf{Q}_i \mathbf{b}_{\mathbf{x},\beta}, v_i : \mathcal{X} \to \mathbb{R}$*, and can be viewed as a scalar Lyapunov function. The parameters matrix,* $\mathbf{Q} \in \mathbb{S}^{\beta n^2 + n}$*, is a block-diagonal of all* $\mathbf{Q}_i \in \mathbb{S}^{\beta n+1}$ *matrices.*

Definition 4.2 introduces a non-conventional LPF candidate. Rather than considering a single LPF, we designate a distinct polynomial LPF for each dimension of the state space and stack them into $v(\mathbf{x}; \ \mathbf{Q})$. This characterization, known as a vector Lyapunov function [37], is less restrictive and enables the policy and LPF to be learned moreindependently for each dimension of the state space.

We highlight that the GAS of the policy in each dimension, $\hat{f}_i(\mathbf{x}; \ \mathbf{P}_i)$, implies the GAS of the entire policy, $\hat{f}(\mathbf{x}; \ \mathbf{P})$. Proposition 4.3 establishes a link between the policy stability in each row and the global stability of the multidimensional policy.

**Proposition 4.3.** *Assuming each pair* $(\hat{f}_i(\mathbf{x}; \ \mathbf{P}_i), \ v_i(\mathbf{x}; \ \mathbf{Q}_i))$ *individually satisfies the GAS conditions. Then, the sum* $\hat{v} = \sum_{i=1}^n v_i(\mathbf{x}; \ \mathbf{Q}_i)$ *yields a valid standard Lyapunov function for* $\hat{f}(\mathbf{x}; \ \mathbf{P})$*, proving that the policy satisfies GAS conditions. The proof is given in Appendix A.1.*

The formulation of the policy and the LPF as multidimensional polynomials empowers us to leverage tools from sum-of-squares (SOS) [15, 38]. The SOS approach boils the Lyapunov GAS conditions down to verifying positive-definiteness of a set of specified matrices. The next two lemmas illustrate the SOS formulation of Lyapunov stability conditions.

**Lemma 4.4.** *The first Lyapunov stability criterion, $v_i(\mathbf{x};\ \mathbf{Q}_i) \succeq 0$, is satisfied for each $i \in \{1, \ldots, n\}$ if $\mathbf{Q}_i \succeq 0$ and $\mathbf{Q}_i \in \mathbb{S}^{\beta n+1}$. The proof is outlined in Appendix A.2.*

**Lemma 4.5.** *The second Lyapunov criterion, $\frac{\partial}{\partial t} v_i(\mathbf{x};\ \mathbf{Q}_i) \prec 0$, is fulfilled for each $i \in \{1, \ldots, n\}$ if there exists a symmetric matrix $\mathbf{G}_i \prec 0$ and $\mathbf{G}_i \in \mathbb{S}^{(\alpha+\beta)n+1}$ such that:*

$$\frac{\partial}{\partial t} v_i(\mathbf{x};\ \mathbf{Q}_i) = \frac{\partial v_i(\mathbf{x};\ \mathbf{Q}_i)}{\partial \mathbf{x}} \frac{\partial \mathbf{x}}{\partial t} = \frac{\partial v_i(\mathbf{x};\ \mathbf{Q}_i)}{\partial \mathbf{x}} \hat{f}(\mathbf{x};\ \mathbf{P}) = \mathbf{b}_{\mathbf{x},\alpha+\beta}^T \mathbf{G}_i \mathbf{b}_{\mathbf{x},\alpha+\beta}, \tag{5}$$

*where $\alpha + \beta$ is the basis degree. The matrix $\mathbf{G}_i$ is acquired by polynomial coefficient matching, and depends on $\mathbf{P}$ and $\mathbf{Q}_i$. We summarize this dependence with the function $\mathcal{G}(\mathbf{P}, \mathbf{Q}) = \mathbf{G}$, where $\mathbf{G}$ symbolizes the block-diagonal form of all $\mathbf{G}_i$ matrices. The proof is outlined in Appendix A.3.*

Finally, with the necessary tools at our disposal, we can establish the connection between the global stability of the policy and finding SOS polynomials in Theorem 4.6. This theorem serves as the fundamental basis for the subsequent policy optimization process.

**Theorem 4.6.** *A polynomial DS policy, $\hat{f}(\mathbf{x};\ \mathbf{P})$, is GAS if the following conditions are satisfied:*

$$(a)\ \mathbf{Q} \succeq 0, \qquad (b)\ \mathbf{G} \prec 0, \qquad (c)\ \mathcal{G}(\mathbf{P}, \mathbf{Q}) = \mathbf{G}. \tag{6}$$

*The proof is straightforward and is sketched in Appendix A.4.*

### 4.3 Joint optimization problem

At this stage, we have established polynomial representations for both the policy and the LPF, along with a firm connection that confirms global stability. Now, we develop an objective function using the Mean-Squared Error (MSE) cost with the Elastic Net Regularization [39]. The MSE is calculated between the policy output and the expert's actions across demonstrated trajectories, and it solely depends on the policy parameters. Essentially, this problem entails regularized polynomial regression to minimize the imitation MSE to expert's demonstrations, subject to the existence of an LPF that satisfies the Lyapunov conditions. The optimization problem is framed as:

$$\min_{\mathbf{Q},\mathbf{G},\mathbf{P}} J(\mathbf{P}) = \frac{1}{2N_t} \sum_{d=1}^{N_d} \sum_{s=1}^{N_s} (\hat{f}(\mathbf{x}^d(s);\ \mathbf{P}) - \dot{\mathbf{x}}^d(s))^2 + \lambda_1 \|\mathbf{P}\|_1 + \lambda_2 \|\mathbf{P}\|_F^2, \tag{7}$$

$$s.t. \quad (a)\ \mathbf{Q} \succeq 0 \quad (b)\ \mathbf{G} \prec 0 \quad (c)\ \mathcal{G}(\mathbf{P}, \mathbf{Q}) = \mathbf{G} \quad (d)\ \mathbf{Q} = \mathbf{Q}^T,\ \mathbf{G} = \mathbf{G}^T,\ \mathbf{P} = \mathbf{P}^T,$$

where $\|.\|_1$ and $\|.\|_F^2$ denote the first and Frobenius norms, and $\lambda_1,\ \lambda_2 \in \mathbb{R}^+$ represent the regularization coefficients. Equation 7 is a semi-definite programming with nonlinear cost function [40, 38], and can be solved using standard semi-definite solvers [41, 42]. Semi-definite programming facilitates optimization over the convex cone of symmetric, positive semi-definite matrices or its affine subsets. Note that $(c)$ can cause the optimization to become non-convex. To alleviate this, we employ SDP relaxations [12], iterative methods based on an initial guess of the $Q$ matrix, and ultimately sequential quadratic programming (SQP) [43].

Notice that the negative-definite constraints can be transformed to a semi-definite constraints by a constant shift. Furthermore, the Lyapunov conditions restrict the gradient to nonzero values away from the origin. The restriction ensures that the LPF has only one global minimum at the target.

## 5 Experiments

We employ two motion planning datasets for experiments. Our primary data comes from the widely recognized LASA Handwriting Motion Dataset [44], which comprises data recorded from handwritten trajectories. The second dataset contains expert demonstrations collected through teleoperating a robotic arm on realistic manipulation tasks. Details about both datasets can be found in Appendix B.1.

### 5.1 Evaluation

For evaluation purposes, we apply PLYDS and the baselines to the dataset (Figure 2a) and evaluate the performance of policy rollouts in PyBullet simulation (Figure 2b) before deploying safe policies

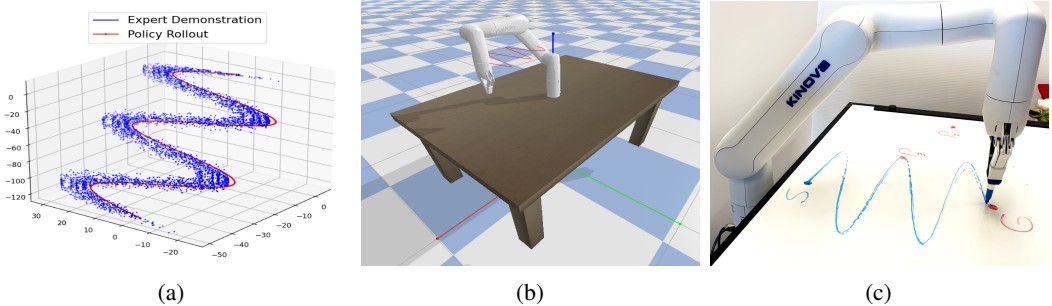

(a)                                      (b)                                      (c)

Figure 2: Overview of the evaluation sequence: (a) learning from demonstrated data, (b) numerical evaluation in simulation and (c) deployed in real-world Gen3 Lite manipulator.

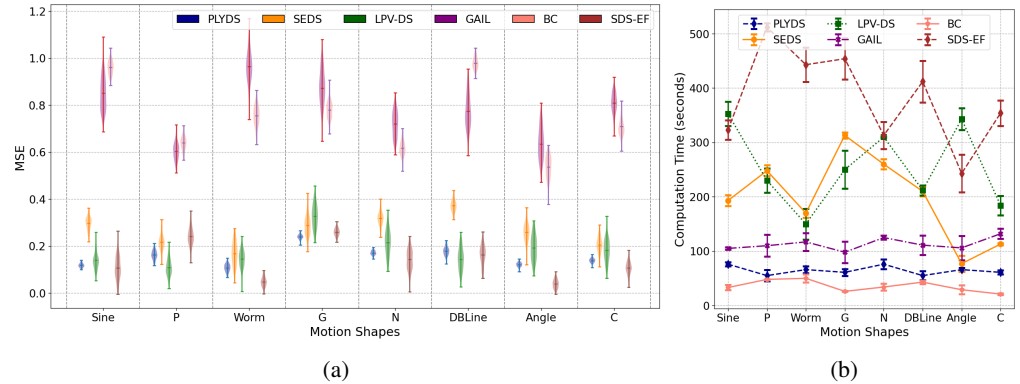

(a)                                                              (b)

Figure 3: Comparison of (a) mean and standard deviation of reproduction MSE and (b) computation time to designated imitation learning methods. PLYDS performs reasonably well in terms of accuracy and is even more promising in terms of computational cost.

onto a manipulator (Figure 2c). In all experiments, we randomly split the demonstrated trajectories in the dataset into train and test sets. The policy learning stage, introduced in Equation (7), is carried out on the training data. The learned policy is subsequently evaluated by calculating the MSE between the policy predictions and the ground truth in the test data, $\frac{1}{2N_d^{test}N_s}\sum_{d=1}^{N_d^{test}}\sum_{s=1}^{N_s}(\hat{f}(\mathbf{x}^d(s);\ \mathbf{P}) - \dot{\mathbf{x}}^d(s))^2$. Recall that the policy output, $\dot{\mathbf{x}}$, is the velocity passed to the robot's low-level controller. We repeat this procedure over 20 different random seeds, and report the average and standard deviation.

We compare the accuracy of our approach to existing baselines. Primarily, we compare against Stable Estimator of Dynamical Systems (SEDS) [3], Linear Parameter Varying Dynamical Systems (LPV-DS) [22], and Stable Dynamical System learning using Euclideanizing Flows (SDS-EF) as methods that ensure GAS. We also compare our method to Behavioral Cloning (BC) [26], and Generative Adversarial Imitation Learning (GAIL) [23] to highlight the importance of global stability. Note that among these, BC and GAIL do not provide mathematical stability guarantees, but the results could provide further comparison ground for the accuracy and computation time. The implementation details, hyperparameters and architecture are discussed in Appendix B.2 and Appendix B.3.

## 5.2 Handwriting dataset

We compare the learned policies of PLYDS to the baselines on eight designated motions. The outcome of these experiments is reported in Figure 3. Despite stability guarantees, the overall accuracy is better among stable imitation learning methods compared to unstable neural approaches.

To analyze GAS, we visualize the learned policies of all methods as streamlines. Figure 4 illustrates the policy rollouts for N-Shaped motion of the handwriting dataset. Each sub-figure represents a trained policy illustrated with gray streamlines. It is evident that SEDS and PLYDS maintain GAS,

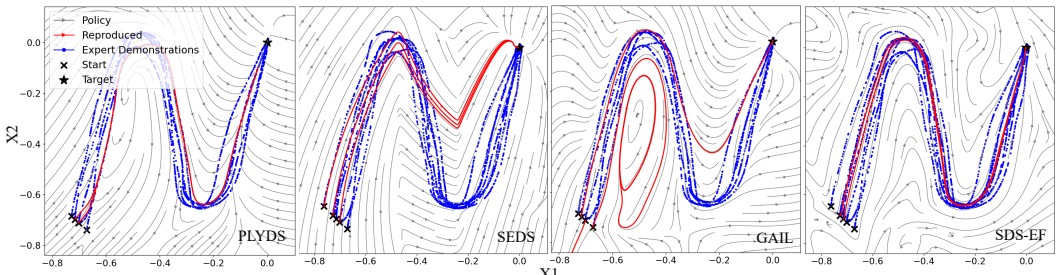

Figure 4: Policy rollout for N-Shaped motion of the handwriting dataset. Each figure represents a trained policy (gray) and rollouts (red) learned from demonstrations (blue). Note the stability issues with GAIL and SDS-EF, where some streamlines fail to reliably converge to the target.

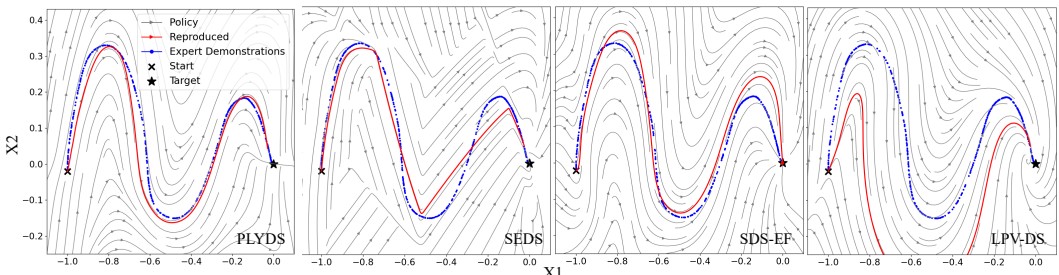

Figure 5: Policy rollout for Sine-Shaped motion (blue) of the handwriting dataset, with access to only *one* expert demonstration. Each figure represents a trained policy (gray) and one rollout (red) learned from one demonstration (blue). Methods requiring large datasets for clustering, such as SEDS and LPV-DS, exhibit inaccurate and unsteady performance.

while GAIL and SDS-EF fail to demonstrate converging trajectories for the entire state space. The same pattern persists for other motions as depicted in Appendix C.1.

Finally, we examine the sample efficiency of our method by reducing the input data to **one** demonstrated trajectory. From Figure 5, we can see that PLYDS learns a stable policy with such limited training samples, while the baselines generate trajectories which diverges from expert data.

So far in this section, the policy and the LPF polynomial degrees were set to $\alpha = 6$ and $\beta = 2$. To understand the way in which the complexity of polynomials affects the overall performance, we repeated the same experiments with degrees of $\alpha = 4, 6,$ and $8$, and present the result in Appendix C.2. We observe that a higher complexity leads to improved precision, if not halted by overfitting or stability sacrifice. Moreover, we study different LPF complexities in Appendix C.3, evaluate the performance of PLYDS with noisy demonstrations in Appendix C.4, and further investigate the computational times in Appendix C.5.

### 5.3 Manipulation tasks

To conduct real-world trials, we collect a second set of expert demonstrations through teleoperating Kinova Gen3 Lite, a manipulator arm with six degrees of freedom. This new dataset holds three distinct motions: (a) root-parabola, (b) standard pick and place, and (c) prolonged-sine, which represent exemplary nonlinear trajectories (Figure 6). Additional details are available in Appendix B.

The performance of all methods is summarized in Table 1, where PLYDS often outperforms other baselines. Next, the learned policy of PLYDS is transferred to the physical arm (Figure 6) and successfully imitates the introduced manipulation tasks. We also start the robot at regions that are further away from the demonstrations and introduce perturbations by randomly pushing the robot to reveal the inherent GAS of PLYDS. As expected, PLYDS manage to successfully recover to the goal.

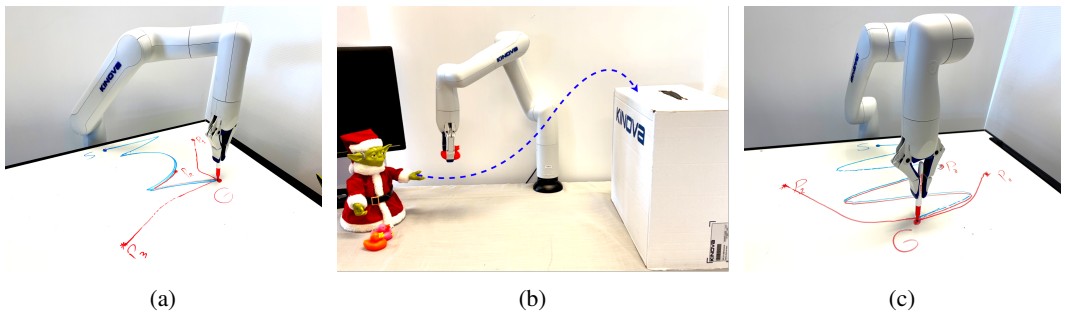

|  (a)  |  (b)  |  (c)  |

Figure 6: Manipulation tasks: (a) root-parabola, (b) standard pick and place, and (c) prolonged-sine.

| Expert Motion | Prolonged Sine | Root Parabola | Pick-and-Place | Computational Time |
|---|---|---|---|---|
| SEDS [3] | $0.234 \pm 0.015$ | $0.152 \pm 0.023$ | $0.094 \pm 0.012$ | $277.02 \pm 13.60$ |
| BC [26] | $1.650 \pm 0.133$ | $0.931 \pm 0.078$ | $0.725 \pm 0.133$ | $38.93 \pm 9.11$ |
| GAIL [23] | $2.322 \pm 0.098$ | $1.322 \pm 0.094$ | $0.663 \pm 0.098$ | $143.15 \pm 8.68$ |
| SDS-EF [8] | $0.234 \pm 0.015$ | $0.152 \pm 0.023$ | $0.094 \pm 0.012$ | $715.62 \pm 18.79$ |
| LPV-DS + P-QLF [22] | $0.234 \pm 0.015$ | $0.152 \pm 0.023$ | $0.094 \pm 0.012$ | $334.55 \pm 25.74$ |
| PLYDS (ours) | $0.111 \pm 0.007$ | $0.176 \pm 0.015$ | $0.021 \pm 0.003$ | $21.37 \pm 1.52$ |

Table 1: Policy rollout reproduction MSE and computational time in PyBullet.

# 6   Conclusion and Limitations

We introduced an approach that aims to learn globally stable nonlinear policies represented by polynomial dynamical systems. We employ the learned policies for motion planning based on imitating expert demonstrations. Our approach jointly learns a polynomial policy along with a parametric Lyapunov candidate that verifies global asymptotic stability by design. The resulting DS is utilized as a motion planning policy, guiding robots to stably imitate the expert's behavior. A comprehensive experimental evaluation is presented in real-world and simulation, where the method is compared against prominent imitation learning baselines.

**Limitations.** A limitation of SOS is that the set of non-negative polynomials is larger than the ones expressed as SOS [45]. Though rare in motion planning tasks, this implies that finding a Lyapunov candidate could be difficult, especially with simultaneous search for a suitable dynamical system. Lasserre hierarchy and SOS extensions [46] can search in a broader class of LPF candidates and tackle this issue. Another limitation occurs when finding highly complex policies that lead to a violation of stability guarantees. This often happens when the regularization coefficients or the optimization tolerance are not set properly. We discuss this trade-off between stability and accuracy in Appendix C.2. Further, the computation complexity of PLYDS is feasible with a reasonable choice of polynomial degrees. Higher degrees are computationally demanding, but are often unnecessary in normal motion planning tasks.

**Future work.** Future work includes incorporating more elaborate safety criteria, such as control barrier functions [47] or real-time obstacle avoidance, into our learning objectives. Plus, applications of our method in SE(3) planning, or other higher-dimensional spaces, such as configuration space of manipulator robots, may be further investigated. Vector Lyapunov functions and adaptable complexity of polynomials can pave the way for such applications, as they assuage major computational challenges.

# 7   Video, Codebase, and Reproducibility

The codebase, video supplements, etc. related to this project are available on our Git repository [1]. Reproducing the experiments is as straightforward as installing dependent software packages, and running a Unix commands in README files.

---

[1] github.com/aminabyaneh/stable-imitation-policy

**Acknowledgments**

This work is sponsored by NSERC Discovery Grant. We also appreciate the thoughtful reviewers' comments which helped us enhance the paper, particularly the experiments.

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

# A   Mathematical Proofs

Due to space limitations and to maintain coherency, proofs for propositions, lemmas, and theorems are presented in this section in the order in which they appeared in the main text.

## A.1   Proof of Proposition 4.3

*Assuming each pair $(\hat{f}_i(\mathbf{x};\ \mathbf{P}_i),\ v_i(\mathbf{x};\ \mathbf{Q}_i))$ individually satisfies the GAS conditions. Then, the sum $\hat{v} = \sum_{i=1}^{n} v_i(\mathbf{x};\ \mathbf{Q}_i)$ yields a valid standard Lyapunov function for $\hat{f}(\mathbf{x};\ \mathbf{P})$, proving that the policy satisfies GAS conditions.*

**Proof.**  As $v_i(\mathbf{x};\ \mathbf{Q}_i)$ is an LPF candidate for $\hat{f}_i(\mathbf{x};\ \mathbf{P}_i)$, both the first and second Lyapunov conditions must be satisfied, i.e., $\forall i \in \{1, \ldots, n\}$:

$$(a)\ v_i(\mathbf{x};\ \mathbf{Q}_i) \succeq 0,\ \forall \mathbf{x} \in \mathcal{X}, \qquad (b)\ \frac{\partial v_i(\mathbf{x};\ \mathbf{Q}_i)}{\partial t} \prec 0,\ \forall \mathbf{x} \in \mathcal{X}.$$

Define the sum of elements $\hat{v}(\mathbf{x};\ \mathbf{Q}) = \sum_{i=1}^{n} v_i(\mathbf{x};\ \mathbf{Q}_i)$. We show that $\hat{v}(\mathbf{x};\ \mathbf{Q})$ satisfies both Lyapunov global stability conditions:

$$(i)\ \ v_i(\mathbf{x};\ \mathbf{Q}_i) \succeq 0\ (a) \Rightarrow v_1(\mathbf{x};\ \mathbf{Q}_1) + \ldots + v_n(\mathbf{x};\ \mathbf{Q}_n) = \hat{v}(\mathbf{x};\ \mathbf{Q}) \succeq 0,$$

$$(ii)\ \ \frac{\partial \hat{v}(\mathbf{x};\ \mathbf{Q})}{\partial t} = \frac{\partial \sum_{i=1}^{n} v_i(\mathbf{x};\ \mathbf{Q}_i)}{\partial t} = \sum_{i=1}^{n} \frac{\partial v_i(\mathbf{x};\ \mathbf{Q}_i)}{\partial t},\ \ \frac{\partial v_i(\mathbf{x};\ \mathbf{Q}_i)}{\partial t} \prec 0\ (b).\ \ \square$$

## A.2   Proof of Lemma 4.4

*The first Lyapunov stability criterion, $v_i(\mathbf{x};\ \mathbf{Q}_i) \succeq 0$, is satisfied for each $i \in \{1, \ldots, n\}$ if $\mathbf{Q}_i \succeq 0$ and $\mathbf{Q}_i \in \mathbb{S}^{\beta n + 1}$.*

**Proof.**  Considering that $v_i(\mathbf{x};\ \mathbf{Q}_i) = \mathbf{b}_{\mathbf{x},\beta}^T \mathbf{Q}_i \mathbf{b}_{\mathbf{x},\beta}$ and $\mathbf{Q}_i$ is not singular, we can perform a Cholesky factorization on the parameters' matrix $\mathbf{Q}_i$. The result is $\mathbf{Q}_i = L_i^T L_i$, and the positivity of $v_i(\mathbf{x};\ \mathbf{Q}_i)$ comes from,

$$v_i(\mathbf{x};\ \mathbf{Q}_i) = \mathbf{b}_{\mathbf{x},\beta}^T \mathbf{Q}_i \mathbf{b}_{\mathbf{x},\beta} = \mathbf{b}_{\mathbf{x},\beta}^T L_i^T L_i \mathbf{b}_{\mathbf{x},\beta} = (L_i \mathbf{b}_{\mathbf{x},\beta})^T (L_i \mathbf{b}_{\mathbf{x},\beta}) = ||L_i \mathbf{b}_{\mathbf{x},\beta}||^2 \succeq 0,$$

that represents $v_i(\mathbf{x};\ \mathbf{Q}_i)$ as an SOS and therefore achieves the first Lyapunov condition.  $\square$

## A.3   Proof of Lemma 4.5

*The second Lyapunov criterion, $\frac{\partial}{\partial t} v_i(\mathbf{x};\ \mathbf{Q}_i) \prec 0$, is fulfilled for each $i \in \{1, \ldots, n\}$ if there exists a symmetric matrix $\mathbf{G}_i \prec 0$ and $\mathbf{G}_i \in \mathbb{S}^{(\alpha+\beta)n+1}$ such that:*

$$\frac{\partial}{\partial t} v_i(\mathbf{x};\ \mathbf{Q}_i) = \frac{\partial v_i(\mathbf{x};\ \mathbf{Q}_i)}{\partial \mathbf{x}} \frac{\partial \mathbf{x}}{\partial t} = \frac{\partial v_i(\mathbf{x};\ \mathbf{Q}_i)}{\partial \mathbf{x}} \hat{f}(\mathbf{x};\ \mathbf{P}) = \mathbf{b}_{\mathbf{x},\alpha+\beta}^T \mathbf{G}_i \mathbf{b}_{\mathbf{x},\alpha+\beta}, \qquad (8)$$

*where $\alpha + \beta$ is the basis degree. The matrix $\mathbf{G}_i$ is acquired by polynomial coefficient matching, and depends on $\mathbf{P}$ and $\mathbf{Q}_i$. We summarize this dependence for all $i \in \{1, \ldots, n\}$ with the function $\mathcal{G}(\mathbf{P}, \mathbf{Q}) = \mathbf{G}$, where $\mathbf{G}$ symbolizes the block-diagonal form of all $\mathbf{G}_i$ matrices.*

**Proof.**  We know that LPF rows are denoted by $v_i(\mathbf{x};\ \mathbf{Q}_i) = \mathbf{b}_{\mathbf{x},\beta}^T \mathbf{Q}_i \mathbf{b}_{\mathbf{x},\beta}$. Hence, we write the second Lyapunov condition by taking the derivative of each row:

$$\frac{\partial v_i(\mathbf{x};\ \mathbf{Q}_i)}{\partial t} = \frac{\partial v_i(\mathbf{x};\ \mathbf{Q}_i)}{\partial x_1} \frac{\partial x_1}{\partial t} + \frac{\partial v_i(\mathbf{x};\ \mathbf{Q}_i)}{\partial x_2} \frac{\partial x_2}{\partial t} + \ldots + \frac{\partial v_i(\mathbf{x};\ \mathbf{Q}_i)}{\partial x_n} \frac{\partial x_n}{\partial t}$$

$$\frac{\partial x_j}{\partial t} = \hat{f}_j(\mathbf{x};\ \mathbf{P}_j) \Rightarrow \frac{\partial v_i(\mathbf{x};\ \mathbf{Q}_i)}{\partial t} = \sum_{j=1}^{n} \frac{\partial v_i(\mathbf{x};\ \mathbf{Q}_i)}{\partial x_j} \hat{f}_j(\mathbf{x};\ \mathbf{P}_j)$$

$$= \sum_{j=1}^{n} \frac{\partial [\mathbf{b}_{\mathbf{x},\beta}^T \mathbf{Q}_i \mathbf{b}_{\mathbf{x},\beta}]}{\partial x_j} [\mathbf{b}_{\mathbf{x},\alpha}^T \mathbf{P}_j \mathbf{b}_{\mathbf{x},\alpha}]$$

Within the last summation, both the derivative of Lyapunov function and the policy are polynomials. The idea is that their multiplication could also be written as an SOS polynomial if the parameters $\mathbf{P}_i$ and $\mathbf{Q}_i$ are chosen carefully. For this polynomial, we define a new basis $\mathbf{b}_{\mathbf{x},\alpha+\beta}$ and $\mathbf{G}_i \in \mathbb{S}^{(\alpha+\beta)n+1}$. Note that the degree of this basis is calculated by $\frac{1}{2}[(2\beta - 1) + (2\alpha) + 1]$, which is the rounded-up degree of the above multiplication term.

Next, we match polynomial coefficients on both sides, yielding $\mathbf{G}_i$ parameters as a function of both $\mathbf{P}$ and $\mathbf{G}_i$, i.e.,

$$\mathbf{b}_{\mathbf{x},\alpha+\beta}^T \mathbf{G}_i \mathbf{b}_{\mathbf{x},\alpha+\beta} \xleftrightarrow[Coefficients]{Matching} \sum_{j=1}^{n} \frac{\partial[\mathbf{b}_{\mathbf{x},\beta}^T \mathbf{Q}_i \mathbf{b}_{\mathbf{x},\beta}]}{\partial x_j}[\mathbf{b}_{\mathbf{x},\alpha}^T \mathbf{P}_j \mathbf{b}_{\mathbf{x},\alpha}]$$

$$\Rightarrow \mathbf{G}_i = \mathcal{G}_i(\mathbf{P}, \mathbf{Q}_i)$$

We summarize the same relationship for all $\mathbf{G}_i$ matrices, and call the resulting function $\mathcal{G}$. Hence, the second condition can be represented by $\mathbf{G} = \mathcal{G}(\mathbf{P}, \mathbf{Q})$ and $\mathbf{G} \prec 0$, and be viewed as SOS.     $\square$

### A.4    Proof of Theorem 4.6

Assuming the polynomial representation of a nonlinear autonomous dynamical system (Definition 4.1),

$$\hat{f}(\mathbf{x};\ \mathbf{P}) = [\mathbf{b}_{\mathbf{x},\alpha}^T \mathbf{P}_1 \mathbf{b}_{\mathbf{x},\alpha}\ \ \mathbf{b}_{\mathbf{x},\alpha}^T \mathbf{P}_2 \mathbf{b}_{\mathbf{x},\alpha}\ ...\ \mathbf{b}_{\mathbf{x},\alpha}^T \mathbf{P}_n \mathbf{b}_{\mathbf{x},\alpha}]^T,$$

the existence of a corresponding polynomial Lyapunov function (Definition 4.2),

$$v(\mathbf{x};\ \mathbf{Q}) = [\mathbf{b}_{\mathbf{x},\beta}^T \mathbf{Q}_1 \mathbf{b}_{\mathbf{x},\beta}\ \ \mathbf{b}_{\mathbf{x},\beta}^T \mathbf{Q}_2 \mathbf{b}_{\mathbf{x},\beta}\ ...\ \mathbf{b}_{\mathbf{x},\beta}^T \mathbf{Q}_n \mathbf{b}_{\mathbf{x},\beta}]^T$$

guarantees the asymptotic global stability of the policy, if the following conditions are satisfied:

$$(a)\ \mathbf{Q} \succeq 0, \qquad (b)\ \mathbf{G} \prec 0, \qquad (c)\ \mathcal{G}(\mathbf{P}, \mathbf{Q}) = \mathbf{G}.$$

**Proof.** The proof is straightforward and follows both Lemma 4.4 and Lemma 4.5. We know that each partial DS $\hat{f}_i(\mathbf{x};\ \mathbf{P}_i)$ is stable if the corresponding parameterized LPF satisfies (a), (b), and (c), where $\mathcal{G}$ is an affine function found in Lemma 4.5 by polynomial coefficient matching. Since each $\hat{f}_i(\mathbf{x};\ \mathbf{P}_i)$ explains the derivative along one of the orthogonal basis of $\hat{f}(\mathbf{x};\ \mathbf{P})$, their individual global stability is equivalent to the stability of the entire system. In other words,

$$\forall x_i \in \mathcal{D}\{\hat{f}_i(\mathbf{x};\ \mathbf{P}_i)\},\ \lim_{t\to\infty} x_i = x_i^*$$

$$\Rightarrow \lim_{t\to\infty} \mathbf{x} = [\lim_{t\to\infty} x_1\ \ \lim_{t\to\infty} x_2\ ...\ \lim_{t\to\infty} x_n]^T = \mathbf{x}^*$$

Another proof can be provided using the LPF introduced in Proposition 4.3 as a Lyapunov candidate for the whole system. Both proofs equally validate the stability of the polynomial DS.     $\square$

## B    Experiment Setup and Details

Enclosed in this section are detailed descriptions of our experiment setup, main software packages, and datasets. Due to space limitations, crucial details from the experiments are explained here. Even though reading the section is not necessary to understand the paper, it provides useful insight into our setup and can aid reproducibility and future research.

### B.1    Datasets

**Handwriting dataset.**    The LASA Handwriting Dataset, partly depicted in Figure 7, is a collection of 2D handwriting motions recorded from a Tablet-PC and by user's input. The dataset includes 30 human handwriting motions, where each motion represents a desired pattern. For each pattern, there are seven demonstrations recorded, with each demonstration starting from a slightly different (but fairly close) initial position but ending at the same final point. These demonstrations may intersect

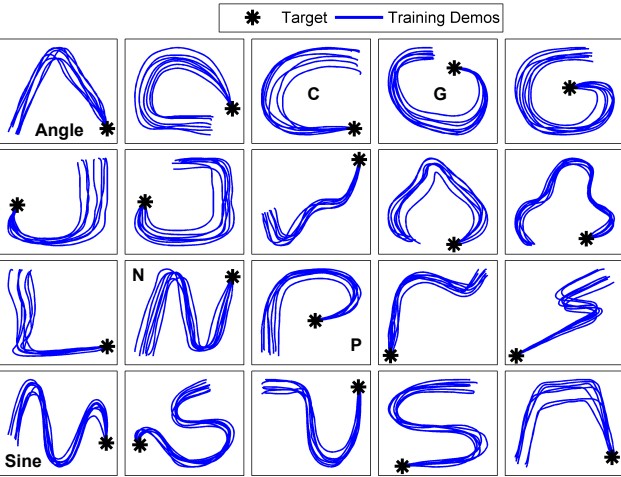

Figure 7: Plots of handwriting dataset motions used in our experiments. We select a representative subset of motions for baselining to keep the experiments computationally feasible. Each plot shows 7 demonstrations with 1000 recorded samples per each. Notice that the time indexing is included in the dataset, but it is irrelevant to our work as we learn time-invariant policies.

with each other. Out of the 30 motions, 26 correspond to a single pattern, while the remaining four motions include multiple patterns, referred to as Multi Models. In all the handwriting motions, the target position is defined as (0, 0), without loss of generality. The dataset provides the following features:

- Position (2 × 1000) representing the motion in 2D space. The first row corresponds to the x-axis, and the second row corresponds to the y-axis in Cartesian coordinates.

- Time (1 × 1000) being the time-stamp for each data point in the motion. We do not use this property, as our proposed method generates time-invariant policies.

- Velocity (2 × 1000) representing the velocity corresponding to each position. We use this feature as a label and form our MSE cost function to calculate the difference between the predicted velocity and this data.

- Acceleration (2 × 1000) matrix representing the acceleration. Not applicable to our research, but could potentially be utilized for future research.

We will not experiment on the entire dataset of 30 motions due to computational unfeasibility. Instead, we select a representative set of motions with (8 × 5 × 2 × 1000) samples in total. The experiments are mainly conducted with this designated set, but since the set is chosen to be representative, we expect the results to generalize to other motions as well.

**Velocity normalization.** Moreover, for some experiments, we opt to normalize the velocity values, such as in Figure 8, to avoid large cost values. This can cause a loss of generality, since the policy actions are now restricted to the direction of the action vector, and will not try to replicate its size. The size of this arrow might be important in scenarios where parts of the motion need to be carried out at a different pace. PLYDS can handle the dataset with or without velocity normalization. The dataset is also referenced and provided as a part of our reproducibility efforts.

**Real-world collected dataset.** We collect data by teleoperating the Kinova Gen3 Lite Arm. Teleoperation involves employing human agents to operate a robotic device or system, recording their actions as expert demonstrations. The teleoperated actions are then recorded and utilized as training data for algorithmic learning. We have two options for teleoperation. First, a human expert can manually control the robot arm using a joystick or keyboard. This process results in natural but non-smooth trajectories. The second approach employs the robot's internal control systems and

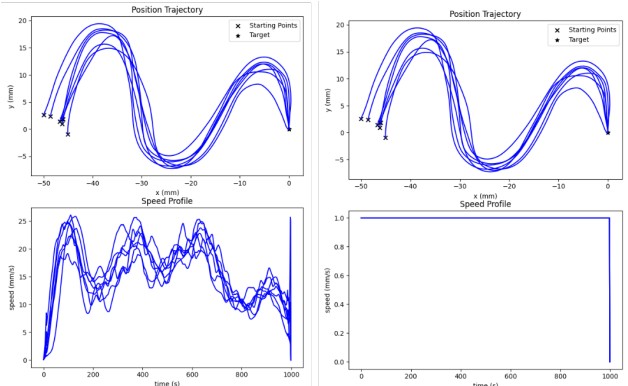

Figure 8: Speed profile for the sine motion, normalized vs. natural. When we normalize the speed, policies fail to capture the difference in the speed vector's magnitude along the trajectories. PLYDS works with both normalized and regular velocities, but we mostly opt for normalized velocities for baselining and comparisons, especially when plotting the policy streamlines for visual verification.

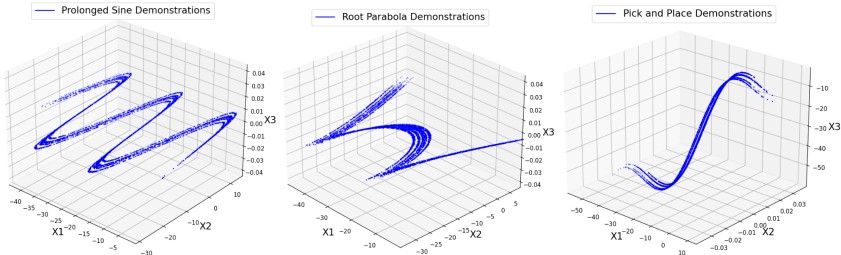

Figure 9: Plots of the dataset collected using Kinova Gen3 Lite and teleoperation. The robot is operated to complete the following trajectories multiple times, while the position and velocity data are recorded in real-time. Expert's demonstrations can also come from robot's low-level controllers, which leads to faster data gathering process. We tried to keep the scale of these trajectories aligned with the handwriting dataset to achieve consistency.

trajectory planning systems to perform as an expert and execute some patterns. This leads to a smoother data collection process with higher reliability. Please keep in mind that planners only connect a series of few points, with no guarantee of stability, and are time-dependent. Consequently, the role of policies generated by PLYDS will not be obsolete.

We gather an open-source dataset holding three distinct motions: the prolonged sine, root parabola, and pick-and-place (Figure 9). Each motion is represented by 50 demonstrations in a 3-dimensional world. Each demonstration contains a state (position) vector ($3 \times 1000$) and a corresponding action (velocity) vector ($3 \times 1000$). Note that orientation is not recorded in the dataset, as we assume the robot's gripper will always face downwards. There will not be any loss of generality because controlling the orientation with PLYDS can be done in parallel, and in the exact same way as the end-effector's position. The dataset is provided as a part of our reproducibility efforts in Section 7.

## B.2 SDP Optimization

We primarily use the commercially available MOSEK [42] optimization software that provides solutions for numerous types of optimization issues, including nonlinear semidefinite programming. The flexibility and high-performance capabilities of MOSEK make it ideal for challenging optimization tasks in both commercial and academic settings. We currently use the MOSEK under an academic license, which can be obtained free of charge with an academic domain email. SCS [48] is another solver specifically designed for solving semidefinite complementarity problems, which include nonlinear SDP as a special case. SCS employs an augmented Lagrangian method combined

with the Fischer-Burmeister function to handle the complementarity conditions in the SDP. At this time, we do not have any solid comparison between the efficiency of these solvers for our setup, but commercial software products often perform more efficient than open-source products.

We also use SciPy [49], an open-source scientific computing library for Python that has many modules for numerical optimization. SciPy can handle a wide range of optimization problems, including nonlinear programming with semidefinite constraints, even though it may not provide specialized solvers for nonlinear SDP. Our software still supports SciPy; however, it is not as efficient in solving nonlinear SDP problems as MOSEK and SCS.

### B.3 Hyperparameters and architecture.

We provide a summary of parameters related to each baseline we used in the paper. Note that we accelerate the computation of GAIL, BC, and SDS-EF with an NVIDIA GeForce RTX 3060 GPU, but SEDS, LPV-DS, and PLYDS use only a Core-i7 Gen8 CPU for optimization. a

**PLYDS.** For our experiments, we primarily utilize parameters $\alpha = 3$ and $\beta = 1$, which have proven effective in most cases. However, we also explore higher degrees to cover a broader range of settings. Additionally, to strike a balance between stability and accuracy, we occasionally adjust the tolerance level from $10^{-4}$ to $10^{-9}$. This allows us to trade off precision for stability when necessary. For the Lyapunov candidate to maintain non-zero gradient beyond the quadratic form, we opt for only square elements in the LPF basis vector. This ensures stability and reliability in our system. Although it is possible to enforce a positive Hessian for the Lyapunov function, it incurs additional computational time while further limiting flexibility in stability conditions. More information about the parameters and architecture of PLYDS can be found on our GitHub repository: github.com/aminabyaneh/stable-imitation-policy.

**GAIL.** The discriminator network takes as input the state-action pairs or observations generated by the policy network and expert demonstrations. *Hidden Layers*: The network may consist of two or three hidden layers, each with 256 or 512 units. *Activation Function*: Rectified Linear Unit (ReLU) activation function is commonly used between the layers. *Output*: The discriminator produces a single output value, representing the probability of the input being from the expert or the generated policy. *Hyperparameters*: Learning Rate: 0.0001, Number of Discriminator Updates per Generator Update: 1 or 2, Discount Factor (for reinforcement learning algorithm): 0.99, Batch Size: 64 or 128, Number of Training Iterations: 1000. We use the imitation package for GAIL's implementation: imitation.readthedocs.io/en/latest/algorithms/gail.html.

**BC.** The behavioral cloning network takes the state-action pairs as input. *Hidden Layers*: The network may have two or three hidden layers, each consisting of 128 or 256 units. *Activation Function*: Rectified Linear Unit (ReLU), *Output*: The output layer of the network corresponds to the action space dimensionality, producing the predicted action. *Hyperparameters*: Learning Rate: 0.0001, Number of Training Iterations: 5000, Batch Size: 64, Regularization Strength (L2 regularization): 0.001, Optimizer: Adam, Loss Function: Mean Squared Error (MSE). Same as with GAIL, we use the imitation package to access BC's implementation: imitation.readthedocs.io/en/latest/algorithms/bc.html.

**SEDS.** Takes position-velocity pairs as input (or state-action pairs in general). *Number of Gaussian Components*: Typically ranges from 3 to 10, depending on the complexity of the motion being learned. *Gaussian Mixture Model* (GMM) Parameters: Covariance Type: Diagonal covariance is commonly used for efficiency and simplicity, Regularization Weight: Often set to a small value, such as 1e-6, to avoid singularities and overfitting, Maximum Number of Iterations: 100 iterations. *Convergence Tolerance*: 1e-6 or 1e-7. We have implemented SEDS in Python using the SciPy optimization library, and the original MATLAB code is not used in our comparisons to remain consistent with other baselines, particularly in terms of computational time. Our implementation of SEDS can be found on github.com/aminabyaneh/stable-imitation-policy.

**LPV-DS.** We mainly use the original implementation of LPV-DS available on: github.com/nbfigueroa/ds-opt and developed in Matlab. The parameters of the method remain the same as the original repository, and we only change the number of demonstrations if required for comparison purposes.

**SDS-EF.** We also provide an implementation of this baseline on our GitHub repository. The coupling layers are set with the following parameters: base network = 'rffn', activation function = 'elu', and sigma=0.45. The main architecture uses 10 blocks and hidden layers' size are set to 200. All these parameters are the same as SDS-EF's original implementation on github.com/mrana6/euclideanizing_flows, but we omit the preprocessing step found in the original implementation, to be able to fairly, and effectively compare the results to other baselines.

## C Supplemental Results

We present a comprehensive set of additional experiments aimed at putting the proposed framework to test from a variety of angles including access to fewer demonstrations, demonstrating the variety of LPFs, more baseline policy rollouts, additive noise, and lastly, we conduct an ablation study by removing the stability guarantee, and present computation times in comparison with the baselines.

### C.1 Baseline policy rollouts

In Figure 10, we plot policy rollouts optimized with PLYDS in comparison to the baselines: SEDS, GAIL, and SDS-EF. We extend these results to LPV-DS in Figure 11. All the portrayed policies are optimized on a set of motions from the handwriting dataset, namely, G-Shaped, Angle, C-Shaped, and P-Shaped demonstrations. The key takeaway is the pattern of instability among neural based imitation learning methods for unknown areas of state space, and inaccuracies visible across the baselines. Hence, the same patterns as the plots in the main text continue to emerge.

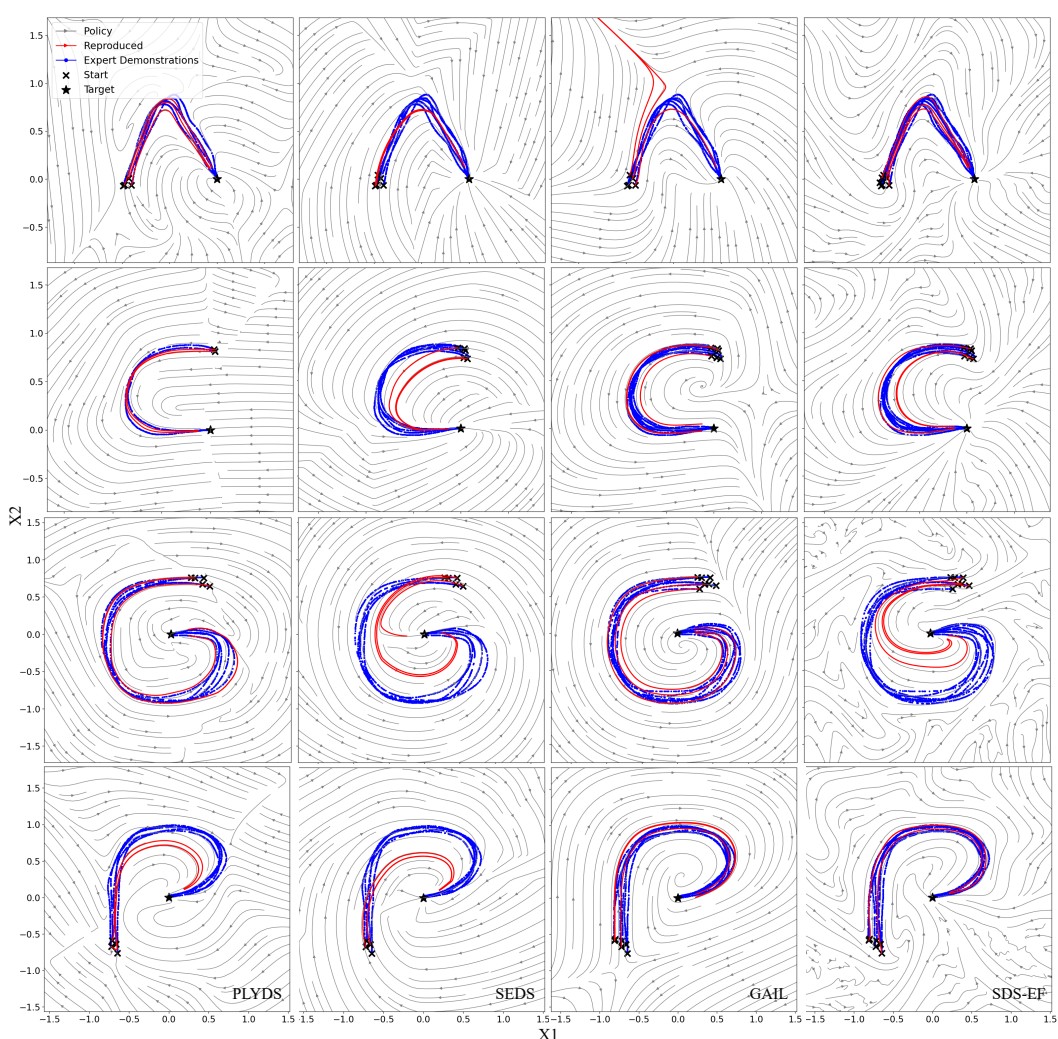

Figure 10: Policy rollout for Angle, C-Shaped, G-Shaped, and P-Shaped demonstrations in hand-writing dataset. PLYDS is visually compared to the baselines in terms of reproduction accuracy and global stability. Note the inaccuracies and unstable reproductions in other baselines.

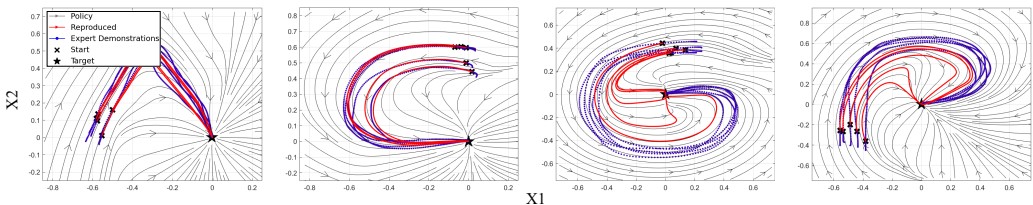

Figure 11: Additional rollouts generated with the LPV-DS's source code in Matlab. These plots serve to complement the results acquired in Figure 10.

## C.2 Ablation study: stability vs. accuracy

When working with DS policies, there is a dilemma known as the stability-accuracy trade-off [22]. This means that a balance must be struck between the reliability and robustness of generated policies to guarantee global convergence to the target (referred to as stability), and minimizing errors to obtain precise solutions (referred to as accuracy). It is important to find a compromise between these two

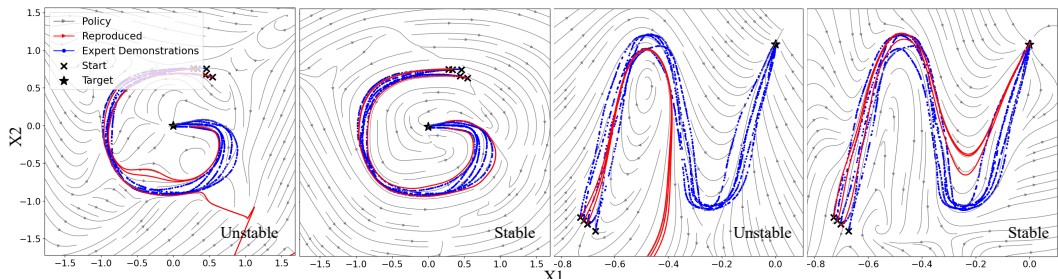

Figure 12: Policy rollouts under utilizing PLYDS, both with and without the imposition of stability constraints, reveal a significant difference in policy behavior. The absence of enforced stability constraints, combined with the utilization of a complex polynomial and a preference for accuracy over stability embedded in the tolerance parameter, results in system instability.

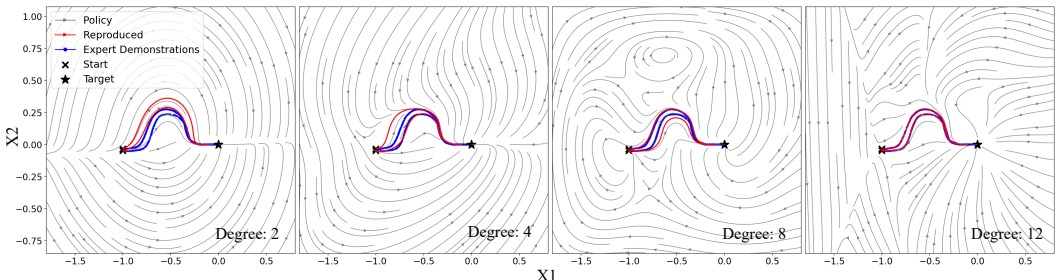

Figure 13: We delve into PLYDS policy rollouts, each employing distinct polynomial complexities. As evident from the plot, the use of increasingly complex polynomials results in more intricate trajectories that better mimic the expert's behavior, but generate more complex trajectories further from the demonstration data. This heightened complexity poses a challenge in ensuring and validating the stability of the system.

factors, as more stable algorithms may not be as accurate, while more accurate algorithms may be sensitive to instabilities.

The higher the degree of the polynomial, the more accurate imitation of expert behavior. Hence, in theory, any nonlinearity may be approximated by our DS formulation. However, in practice, we introduce regularization and tolerance parameters in the code. The tolerance can be used to choose in the favor of accuracy or stability (see Figure 12). Another way to balance this equation is to start with lower-degree polynomials, and increase the policy's complexity when the accuracy is insufficient. Figure 13 serves as an illustration for this process.

### C.3  Complexity of Lyapunov functions

Figure 14 demonstrates the complexity of the Lyapunov function affects trajectory planning in the state space in various ways, such as optimization efficiency, trajectory smoothness, obstacle avoidance, robustness to perturbations, and planning accuracy. If the Lyapunov function is more complex, it may increase computational costs but in turn result in more complex and nonlinear trajectories. On the other hand, simpler Lyapunov functions may offer faster computations, smoother trajectories, and satisfactory planning accuracy, but they may not be adaptable to complex expert demonstrations.

When deciding on the complexity of the Lyapunov function, it is necessary to consider the feasibility of computations, and the desired smoothness and accuracy of planned trajectories and always start with the most simple: quadratic distance function. Figure 14 illustrates this by showing both stable and unstable Lyapunov possibilities gauged across various LPF complexities.

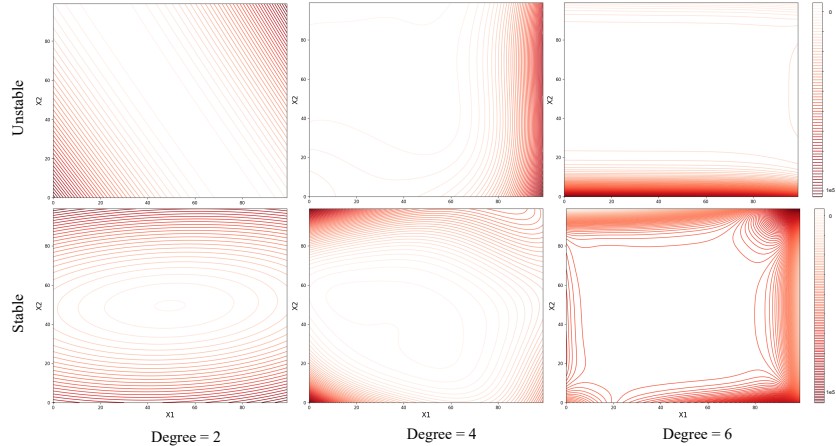

Figure 14: During our policy optimization, we obtain LPF samples such as the ones depicted above. Even though variation in complexity notably affects the computation time, employing more complex Lyapunov functions (polynomials of higher degrees) appears necessary to achieve stable and precise policies in some cases. Currently, we manually determine the complexity of the Lyapunov function and shift to higher complexities only if the optimization fails to deliver satisfactory results.

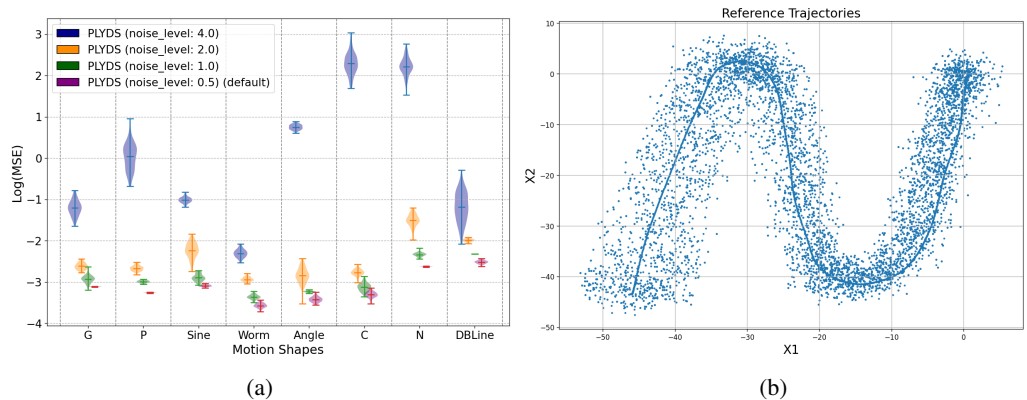

Figure 15: Performance of PLYDS in the face of uniform additive noise (a) and a sample of a noisy trajectory with noise-level set to 2 (b). Noise levels are in centimeters, therefore, a noise-level of 4 means each reading could be deviated from its true value by $\pm 4cm$.

## C.4 Performance with additive noise

Noise in imitation learning significantly impacts the learning process and resulting policies. Excessive noise levels can destabilize the algorithm, preventing it from converging to an optimal policy. To assess PLYDS performance while exposed to noisy measurements, we apply uniform additive noise, distributing samples evenly across a specified interval. We vary the size of this interval, expanding it symmetrically around zero for positions, while also accounting for its effect on velocities within the expert dataset. The results in Figure 15 demonstrate a moderate level of noise-robustness that can be further improved in future studies. Noisy data also increases the error bands, leading to increased uncertainty in the outcome of policy optimization.

## C.5 Computation times

We performed all experiments on a machine equipped with a Core i7 8th Gen CPU, an NVIDIA GeForce RTX 3060 GPU, and 32 GB DDR2 RAM. Among the methods included in our experiments, GAIL, BC, and SDS-EF utilize the GPU to expedite neural network computations. On the other hand,

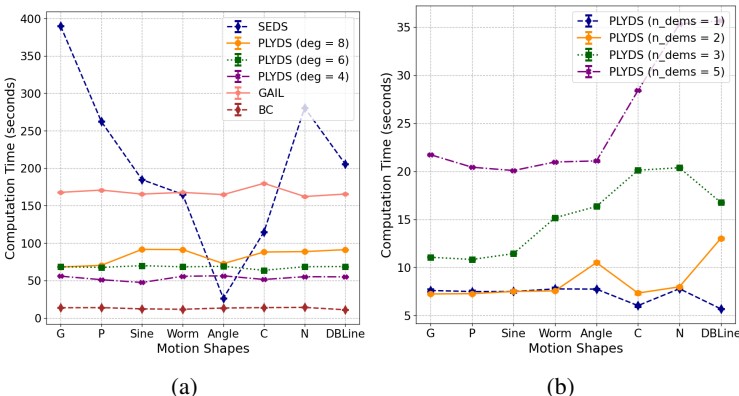

(a)              (b)

Figure 16: Total computation times averaged over 20 trials for PLYDS compared to other baselines (a) and with different dataset sizes (b). It is noteworthy that GAIL and BC are utilizing a GPU to accelerate their processing power.

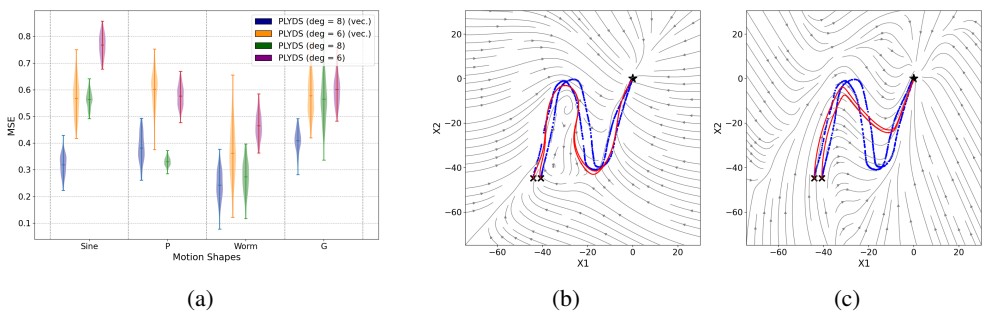

(a)            (b)            (c)

Figure 17: Accuracy (MSE) and policy rollouts for vectorized (left) vs. scalar (right) Lyapunov function. For policy rollout, we picked the N-Shaped motion, where the non-vectorized Lyapunov functions results in a visible reduction in reproduction accuracy. However, the accuracy comparison shows that the extent of improvements caused by vector Lyapunov functions (marked by vec.) depend on the shape of each motion, and may not be verified visually for all motions.

PLYDS, SEDS, and LPV-DS solely rely on the CPU for policy optimization. Despite this variance in computational resources, we provide a comparison of computation times in Figure 16.

## C.6  Ablation study: vector Lyapunov functions

Vector Lyapunov functions [37], extend the concept of Lyapunov stability to systems where state variables are represented as vectors rather than scalars. This extension is particularly valuable when dealing with interconnected or multidimensional systems. Instead of relying on a scalar function, our approach utilizes a vector-valued Lyapunov function to assign a vector to each point in the state space. The properties of this vector-valued function are leveraged to analyze the stability and convergence behavior of system trajectories.

We employ this technique known to enhance the flexibility of the optimization process, as discussed in [50]. Moreover, our observations indicate accuracy improvements over the non-vectorized version for certain motion scenarios. In Figure 17, we present the results of an ablation study focusing on vector Lyapunov functions, highlighting their essential role as a small yet critical component of our method. It is also known that Vectorizing the Lyapunov function yields a higher flexibility during optimization, can potentially lower the computational cost for higher order polynomials.

