# OpenReview forum: "Learning Lyapunov-Stable Polynomial Dynamical Systems Through Imitation"
_robot-learning.org/CoRL/2023/Conference — CoRL 2023 Poster_

### Official Review · Reviewer_ZV92 · 2023-07-01

**Confidence:** 5
**Originality:** Fair
**Technical Quality:** Fair
**Clarity Of Presentation:** Good
**Impact:** 3

**Recommendation:**

Weak Accept: I recommend accepting the paper, but will not argue for my recommendation if the majority of other reviewers have a different opinion.

**Review:**

**Pros:**
1. The paper is written in clear English, and most sections are easy to understand.
2. The experiments are conducted on non-trivial simulation and hardware cases, and the results are impressive.

**Cons:**
1. Do we really need Lyapunov/stability in imitation: the Lyapunov function only ensures the trajectories can finally approach the target point. But it cannot help more in mimicking the expert demonstration (other than reaching the set point), and the "so-called" stability is not that important in this case. One thought to make the method proposed here useful is to consider the stability in the error space - the error for the current state and the corresponding reference state in the expert trajectory. But from the current setup, it is unclear how easy it is to adapt to this "error space" setting.
2. Baselines: In the related work, the author mentioned that neural Lyapunov functions typically lack GAS guarantees. However, the neural Lyapunov methods empirically still work well in high-dimension systems [1,2]. To fully show the advantage of the method proposed in this paper, I suggest comparing the neural Lyapunov methods. In addition, the BC and GAIL don't utilize the goal information as the proposed method and SEDS do, thus, the comparison is a little unfair. I suggest comparing it with [3], which utilizes the additional prior information (the goal point information). A simple comparison would be to enforce an L2 loss for the last several states in the planned trajectories and integrate it with the loss function in Eq (4) in [3]. Also, in terms of neural network structure, a recurrent neural network might be preferable since it captures better the temporal information of the trajectory.

References:
1. Dawson, Charles, et al. "Safe nonlinear control using robust neural lyapunov-barrier functions." Conference on Robot Learning. PMLR, 2022.
2. Chang, Ya-Chien, and Sicun Gao. "Stabilizing neural control using self-learned almost Lyapunov critics." 2021 IEEE International Conference on Robotics and Automation (ICRA). IEEE, 2021.
3. Bhattacharyya, Raunak P., et al. "Simulating emergent properties of human driving behavior using multi-agent reward augmented imitation learning." 2019 International Conference on Robotics and Automation (ICRA). IEEE, 2019.

**Quality Of The Limitations Section:**

Limitations are addressed clearly

**Questions For Rebuttal:**

I hope the authors can address more for the "Cons" section mentioned above. Besides are some extra questions:

1. Why the Polynomial Lyapunov Candidate has to be designed? I read through section 4.2 but was not very clear about it. Since the final Lyapunov function is just a sum of all the Lyapunov candidates, why cannot just learn a Q=\sum\limits_{i=1}^n Q_i? Will it not work empirically?
2. Lines 142-143 mentioned "enables the policy and LPF to be learned independently for each dimension of the state-space" - why those LPF are designed for each dimension, and why are those dimensions in the state space independent? From Eq.(4) it is hard to tell those dimensions are independent, and the state $\mathtt{x}$ shows up in $\mathtt{b_{x,\beta}}$ and every dimension in $v(\mathtt{x}; \mathtt{Q})$ has the component of $\mathtt{b_{x,\beta}}$, thus certainly each dimension in the state space is not independent.
3. Are all the expert trajectories not overlapping each other? Because in the state space, it is often required that from any point (other than the equilibrium point), there should be only one trajectory coming in (out).

**Robotics Focus:**

Sufficient demonstration on hardware

**Summary Of Paper:**

This paper proposes a method to learn stabilizing policy for polynomial dynamical systems using expert demonstrations. It parametrizes the policy via the P matrix and parametrizes the Lyapunov certificate via the Q matrix, and they jointly learn P and Q matrices via joint semi-definite programming (SDP). The optimization problem is solved via a classical SDP solver. They provide theoretical guarantees for stability (provided that the optimization is feasible). They conduct one simulation on handwriting and one hardware experiment for manipulation tasks. They outperform baselines (SEDS, BC, and GAIL) and get better performance for out-of-distribution samples.

**Summary Of Recommendation:**

Overall, I think the experiment has good initial results, but the paper's idea is a bit incremental and it lacks convincing baselines such as neural CLFs and GAIL with prior information for the goal point. Overall my decision is weak reject.

---

### Official Review · Reviewer_fdCw · 2023-07-18

**Confidence:** 5
**Originality:** Good
**Technical Quality:** Good
**Clarity Of Presentation:** Very Good
**Impact:** 4

**Recommendation:**

Weak Accept: I recommend accepting the paper, but will not argue for my recommendation if the majority of other reviewers have a different opinion.

**Review:**

The paper has several strengths, including comprehensive comparisons with various methods and demonstrations on a robotic platform. Additionally, the authors provide the implementation code, which facilitates result reproducibility and enhances the paper's impact.

However, there are a few drawbacks to the method presented in the paper. For instance, the scalability of the method is limited, as it was not tested on the entire LASA dataset due to computational unfeasibility. Furthermore, certain aspects of the paper could benefit from further clarification.


**Quality Of The Limitations Section:**

Additional details required

**Questions For Rebuttal:**

- ε in (2) is not defined

- It is said that (7) is a semi-definite programming with a nonlinear cost function, but constraint (c) of (7) could induce bilinear/equality constraints. When this happens, (7) is not more an SDP in its constraints nor a convex problem.

- No reference is cited for the SCP solver

- It is claimed that higher degrees than the ones used in the authors' experiment are often unnecessary in normal tasks, but in Appendix B.1 is also said that the method was not experimented on the entire dataset due to computational unfeasibility. Please clarify.

- In the Limitations subsections, it said that the Lasserre hierarchy and SOS extensions presented in [43] could search in a broader class of LPF candidates and tackle the issue of finding Lyapunov functions. However, the methods presented in [43] assume that f is given, while in the paper, a simultaneous search needs to be also carried out to find $f$. Thus, in the reviewer's opinion, this does not directly solve the issue. Nevertheless, the authors could check the methods presented in "A Survey of Recent Scalability Improvements for Semidefinite Programming with Applications in Machine Learning, Control, and Robotics" which aims to mitigate the scalability of SOS methods in general.

- The authors write that "A limitation of SOS is that the set of non-negative polynomials is larger than the ones expressed as SOS [42]. Though rare in motion planning tasks, this implies that finding a Lyapunov candidate could be difficult". This implication is not entirely true, as Ahmadi and Parrilo proved in "Converse results on existence of sum of squares Lyapunov functions" that the existence of a polynomial Lyapunov function of a given degree implies the existence of a polynomial Lyapunov function $V$ of (possibly higher degree) that satisfies
the sos conditions $V$ is SOS and $-dV/dt$ is SOS. Nevertheless, the authors are simultaneously searching for \hat{f} and $V$, so the limitation presented in [42] is relevant here. But I think the authors should clarify that the result of [42] implies that simultaneously finding the polynomial system and the Lyapunov function could be difficult.

- It is known that the scalability of vanilla SOS methods is very challenging since the size of the matrices involved in the problem can quickly grow with the degree. This should be more clearly stated in the Limitation subsection.


**Robotics Focus:**

Sufficient demonstration on hardware

**Summary Of Paper:**

The authors propose the use of stable polynomial dynamical systems to encode expert demonstrations in the context of learning from demonstrations.

To guarantee the stability and accuracy of the polynomial dynamical systems, the authors propose an optimization method based in sum-of-squares optimization.

The paper includes a thorough comparison with state-of-the-art methods, and the experiments demonstrate that this method can outperform the state-of-the-art in specific scenarios while also being efficient in terms of sample usage.


**Summary Of Recommendation:**

While polynomial dynamical systems have been extensively studied in the control and systems literature, their application in the context of motion planning has not been explored until now.

This paper fills that gap by connecting polynomial dynamical systems with motion planning, thereby paving the way for further research. By incorporating the substantial body of literature on sum-of-squares (SOS) methods and polynomial dynamical systems into motion planning, this paper makes a valuable contribution to the field of learning from demonstration (LfD) using dynamical systems.

---

> ### Author Response · Authors · 2023-08-11
> **Additional baselines**
>
> In addition to our original response to reviewer's concerns, we present a thorough comparison of our method with more recent baselines, namely:
>
> Asif Rana et al., Euclideanizing Flows: Diffeomorphic Reduction for Learning Stable Dynamical Systems, L4DC’20
>
> and
>
> Figueroa Fernandez NB, Billard A. A physically-consistent Bayesian non-parametric mixture model for dynamical system learning. Proceedings of Machine Learning Research. 2018.
>
> We attached a PDF of these results to the initial response. Please let us know if we can further clarify.

---

### Official Review · Reviewer_x3iz · 2023-07-18

**Confidence:** 5
**Originality:** Fair
**Technical Quality:** Good
**Clarity Of Presentation:** Good
**Impact:** 3

**Recommendation:**

Weak Accept: I recommend accepting the paper, but will not argue for my recommendation if the majority of other reviewers have a different opinion.

**Review:**

Strengths: The paper is well-written, has extensive experiments, and as a contribution in stable learning from demonstration is of relevance to the robot learning community.

Weaknesses and concerns:
- There is a gap in related work and a lack of comparison to modern approaches for learning stable systems with guarantees.
    - The idea of using leveraging polynomials to learn and certify stability of dynamics is not a new one. For instance, consider the work [1] on learning contracting dynamical systems along demonstration trajectories with local certificates of convergence, or [2] for learning Lyapunov stable polynomial latent dynamics in the context of control from pixels.
    - Recent methods leveraging ideas from Riemannian motion policies [3, 4] which learn diffeomorphisms from a known stable policy to a latent space with dynamics that are trained to resemble the demonstrations.
    - I would say the claim that neural Lyapunov methods “typically lack GAS since they softly enforce the Lyapunov stability conditions” in the related work is a bit misleading. Despite using NNs for the dynamics and Lyapunov function, [5] ([33] in the paper) does provide hard stability guarantees (for the learned dynamics) by using counterexample-guided synthesis, and verification using mixed integer programming. This provides hard guarantees at the cost of an expensive verification procedure. On the other end of the spectrum, there are methods leveraging the Lipschitz constants of the Lyapunov/contraction conditions of the learned certificate function and using these to certify violations of the conditions cannot occur over some compact set [6, 7]; these approaches are more conservative in verification but are more scalable than mixed integer verifiers.
- I am confused as to how the optimization in (7) is a (convex) SDP. If one searches for the dynamics (parameterized by P) and the Lyapunov function (parameterized by Q), there should be bilinear terms in the coefficient matching constraint (5), which renders the problem non-convex. There are approaches for solving this iteratively through alternating between fixing P and optimizing Q and vice versa (like in [2]), but it seems this should be a non-convex problem at its core.
- The results would be more convincing if the method is compared with the existing techniques in learning stable polynomial systems from demonstrations (e.g., [1] or [4]).
- The global stability guarantees coming from the SOS program are at odds with the limitations of practical robotic systems, which have bounded domains and control limits. This can compromise the guarantees of the learned behavior. For instance, since the Kinova arm in the experiments has joint limits, there could be the case where the arm is perturbed to some point near the the joint limits where the learned dynamical system drives the arm to exceed the joint limits in an unsafe fashion.
- It would be good to see an ablation on using vector Lyapunov vs. scalar Lyapunov functions. I am unclear on the benefit of having separate Lyapunov functions per dimension and then enforcing that they elementwise all satisfy the standard Lyapunov conditions. It seems that if you can find Q_1, …, Q_n which all satisfy the Lyapunov conditions independently, you should be able to find a single Q for a scalar Lyapunov function (maybe with a polynomial basis of slightly higher degree). I think the idea of using vector Lyapunov functions is interesting, as they admit weaker stability conditions (e.g., component-wise you do not need to be globally negative semidefinite, see [8]), but it seems their advantages are not yet being fully exploited in this work.

[1] El Khadir, Varley, Sindhwani. Teleoperator Imitation with Continuous-time Safety, RSS’19.

[2] Chou and Tedrake. Synthesizing Stable Reduced-Order Visuomotor Policies for Nonlinear Systems via Sums-of-Squares Optimization, CDC’23.

[3] Asif Rana et al., Euclideanizing Flows: Diffeomorphic Reduction for Learning Stable Dynamical Systems, L4DC’20

[4] Zhang, Beik-Mohammadi, and Rozo. Learning Riemannian Stable Dynamical Systems via Diffeomorphisms, CoRL’22.

[5] Dai, Landry, Yang, Pavone, and Tedrake. Lyapunov-stable neural network control, RSS’21.

[6] Chou, Ozay, and Berenson. Model Error Propagation via Learned Contraction Metrics for Safe Feedback Motion Planning of Unknown Systems, CDC’21.

[7] Sun, Jha, and Fan. Learning Certified Control using Contraction Metric, CoRL’20.

[8] Nersesov and Haddad. On the Stability and Control of Nonlinear Dynamical Systems via Vector Lyapunov Functions, TAC’06.

**Quality Of The Limitations Section:**

Additional details required

**Questions For Rebuttal:**

Please see review for specific questions.

**Robotics Focus:**

Sufficient demonstration on hardware

**Summary Of Paper:**

This paper presents a method for learning stable dynamical systems from demonstrations. To obtain hard guarantees on stability, the learned dynamics and Lyapunov stability certificate are represented with polynomials, and an optimization is solved to synthesize the dynamics and Lyapunov function jointly. The method is evaluated on the LASA dataset and on manipulation tasks, outperforming baselines in stability and demonstration tracking.

**Summary Of Recommendation:**

The paper is overall well-written but needs some clarification on the method, expansion of related work, and comparison with some more modern baselines.

---

### Official Review · Reviewer_gRUM · 2023-08-03

**Confidence:** 5
**Originality:** Good
**Technical Quality:** Good
**Clarity Of Presentation:** Very Good
**Impact:** 3

**Recommendation:**

Weak Accept: I recommend accepting the paper, but will not argue for my recommendation if the majority of other reviewers have a different opinion.

**Review:**

**Strengths**
- The paper is well-written and enjoyable to read.
- I went through all the proofs in the appendix and the math seems sound and refreshingly very well explained.
- This approach can be seen as is a very interesting application of "polynomial regression", which is having a resurgence thanks to recent comparisons to NN.
- The transformation of P and Q to G is a very clever trick to solve the problem of learning the parameters of each of the polynomial functions as it allows them to use standard SOS to estimate them.
- I did not have time to test the code, but it looks very well written too.
- Solid solution to a well-known problem.

**Weaknesses**
1. **Motivation/Justification:** Although I appreciate this polynomial formulation and the fact that it allows efficient parameter estimation, I am yet to understand why another SEDS-improvement approach is necessary that only solves for the stability vs. accuracy dilemma. As mentioned by the authors, there are many improvements to SEDS, like the
   - i) the CLF learning approach proposed in 2015  https://www.sciencedirect.com/science/article/abs/pii/S0921889014000372
   - ii) the LPV-DS approach that decouples the GMM parameter estimation to the linear DS estimation stages and uses a P-shaped Lyapunov function already shows superior performance for the type of motions that are being demonstrated in this work (highly non-linear, single-attractor systems) https://proceedings.mlr.press/v87/figueroa18a.html
   - iii) the Euclideanizing-Flow approach that learns a diffeomorphic transformation using normalizing flows https://proceedings.mlr.press/v120/rana20a.html

   I only mention these 3 as they are the ones I know that have implementations/code available offline for the authors to compare.  If the PLYDS approach gives the same performances as these improved (more recent) solutions, then what is the contribution from this new formulation? In general, the majority of the state-of-the-art approaches can already achieve similar reproduction accuracy on the LASA dataset and can handle highly nonlinear trajectories. They all, however, suffer from either computational complexity (during learning and execution) as well as hyper-parameter tuning. So, can the authors clarify or spell out what is the improvement of PLYDS not against SEDS but against the more recent state-of-the-art that already outperform SEDS? This leads me to my next point.

2. **Insufficient Evaluation:**  Using SEDS as a baseline + knowingly unstable policies (BC and GAIL) is not a valid comparison in my opinion. The work presented in this paper is offering an improvement on SEDS, yet many improvements of SEDS exist. It's fine to have SEDS as the lowest baseline approach which we know will not perform well, but to corroborate your claims you should at least compare to one of the improved DS learning approaches to have a more fair performance evaluation. Browsing through your code I saw that SEDS is implemented from scratch in Python. The authors could consider implementing this approach http://proceedings.mlr.press/v78/ravichandar17a/ravichandar17a.pdf which is exactly the same SEDS formulation but with partial contraction constraints instead of Lyapunov. Furthermore, there is no discussion about the computational time in the main text, I see that this is discussed in the appendix. If possible, I suggest to move some of this relevant information to the main text as it's one of important improvement. Can you clarify if for the SEDS computation time is considering the model selection part or is it only timing the SEDS learning algorithm on a predefined number of Gaussians?  This leads to the next point.

3. **Hyper-parameter Tuning:** I feel like the fact that the complexity of the polynomials for both the motion policy and the Lyapunov function are manually defined is not fully addressed in the main text. As you already show in the appendix. the degree of the polynomial affects both reproduction accuracy and stability so a proper set of degrees for each polynomial function must be found, otherwise the claim that PLYDS will outperform SEDS does not hold universally. However, this is not such a pressing issue as very few of the DS learning approaches (with the exception of https://proceedings.mlr.press/v87/figueroa18a.html) requires model selection or hyper-parameter grid search to fund such optimal parameters. Nevertheless, it would be great if the authors discussed or provided some rule-of-thumb for selecting these hyper-parameters or an approach to do such selection automatically or in a data-driven way.

3. **Scalability to higher dimensions:** The experiments in this paper are limited to 2D-3D position trajectories of the robot's end-effector. As I mentioned before, learning stable DS in 2D-3D from highly nonlinear trajectories has already been solved in a myriad of approaches. Yet, apart from the computational limitations many formulations also lack the ability to scale to higher dimensional complex spaces, like $SE(3)$ full end-effector pose or in joint space $q\in R^N$, like RMPs https://arxiv.org/abs/1801.02854. Hence, a more relevant improvement would be to show that this polynomial formulation allows to learn DS in such high-dimensional spaces. Is it possible? How would this affect the required degree of polynomials and computation efficiency, if at all?

**Quality Of The Limitations Section:**

Additional details required

**Questions For Rebuttal:**

Please address weaknesses listed above.

**Robotics Focus:**

Sufficient demonstration on hardware

**Summary Of Paper:**

This paper offers a solution to the stability vs. accuracy dilemma that is the crux of learning reactive motion policies as stable dynamical systems from demonstrations. In this work, this is achieved by formulating both the DS motion policy and the corresponding Lyapunov function (used to enforce stability) as multidimensional polynomial functions. An issue of Lyapunov-based DS learning approaches is the need to estimate both the policy and the Lyapunov functions. In this paper, the parameters of both of these polynomial functions are estimated jointly using SOS by generating a single polynomial representation constructed from a polynomial coefficient matching strategy. This presented approach, named PLYDS, is shown to outperform the first baseline stable DS learning method from 2012 (SEDS) and unstable learning approaches like BC and GAIL, in terms of reproduction accuracy while ensuring global asymptotic stability.

**Summary Of Recommendation:**

The paper presents a novel polynomial dynamical system formulation and efficient learning scheme to tackle the stability vs. accuracy dilemma. The approach is sound, however, the comparisons are insufficient to support the claims. If the authors can address the weaknesses I mention above and particularly show improved performance over any (at least one) of the state-of-the-art approaches that outperform SEDS I would be willing to increase my score.

**Post rebuttal**

The authors have addressed my major concerns regarding comparisons to state-of-the-art. I recommend they highlight these comparisons in the main text. Also, even though it was not addressed in the rebuttal it would be great if the authors could discuss the scalability of their approach to higher dimensional systems and manifolds such as SE(3) and joint space.

---

### Author Response · Authors · 2023-08-14
**Revised Paper**

As promised in earlier responses, we addressed all the major concerns raised by the reviewers where possible, and revised the paper to reflect the invaluable reviewers' suggestions.

The updated paper can be found on [this link to the revised paper](https://drive.google.com/file/d/18s-rSI7Nn25NTGSey9B9s-6uWW6YNseP/view?usp=sharing). The changes are made in blue color for better visualization. The revision of the appendix is currently in progress.

We have dedicated a great deal of time and effort to reflect the reviewer's concerns the best we could, and improve our work further. The reviewers input has indeed greatly contributed to our paper's refinement. We believe that our work has significantly improved during the review process, and we have the reviewers to thank for their insightful comments.

---

### Author Response · Authors · 2023-08-14
**Reviewers' Response**

Dear responsible area chair (AC),

In the past two weeks, we have dedicated a great deal of time and effort to writing author responses, adding and implementing more baselines, revising the paper, and conducting further experiments such as ablation studies to address the reviewers' concerns.

We are, therefore, distressed that despite our timely response (August 7th, only 4 days after the reviews), we have not heard back from the reviewers so far. This is even more important for our work, as most of the scores hover around the borderline.

Please let us know if we should consider adding more material or responses to get the reviewers' attention.

Sincerely,
Authors of Paper274

---

> ### Comment · Area_Chair_W1ES · 2023-08-14
> **RE**
>
> Dear authors,
>
> Unfortunately, I've already contacted the reviewers to get back to you, and I did not recieved any response.
>
> I'll try to contact them again.
>
> CoRL promotes the discussion, but unfortunately reviewers may have different commitments and may not be able to respond in time.
>
> I suggest to add as much material/discussion possible and I'll take this in consideration in the discussion phase.

---

> > ### Author Response · Authors · 2023-08-15
> > **RE: Rebuttal Highlights**
> >
> > Dear responsible AC,
> >
> > Thank you for your efforts and understanding. We realize that reviewers have a busy schedule; yet, in case of no further response, we hereby provide a summary of the rebuttal and highlights of the discussions:
> >
> > * To address reviewers' primary concern, we implemented **two additional recent baselines**, the results are updated both in the paper and in a PDF file replied to reviewers. With these two, we have more baseline comparisons than any other method cited by reviewers gRUM, x3iz, and ZV92. The latter already improved their score but no reply from the other two reviewers so far.
> > * We conducted an **ablation study** as requested by reviewer x3iz.
> > * We revisited our **motivation**, as requested by reviewer gRUM, and focused more on the drawbacks of more recent methods, such as sample inefficiency, computation time, and quasi-stability.
> > * We **updated our original submission** to reflect even the slightest of reviewers' comments where possible, unless, for example, they proposed a baseline not matching the scope of our work, or results that already exist in the paper.
> > * Our work is supported by concrete **mathematical proofs** and comprehensive simulation and **real-world experiments**.
> > * Should there be a need for further adjustments, **we are committed to improve** our paper further to better reflect its quality and contributions.
> >
> > We thank you for taking our endeavor towards addressing reviewers' comments into account. We appreciate reviewer ZV92 for raising their score, and remain optimistic about similar outcomes from other reviewers once they have an opportunity to review our responses.
> >
> > Regards,
> >
> > Authors of Paper274

---

### Decision · Program_Chairs · 2023-08-30

**Decision:**

Accept (Poster)

**Comment:**

This paper tackles the problem of imitation learning under stability guarantees by learning a parametric polynomial planning function. The approach learns the polynomial coefficients jointly with a Lyapunov candidate to ensure stability.

All reviewers appreciated the clarity and the extensive experimental evaluation of the paper, particularly after the author's response that clarified the remaining doubts and presented additional comparisons.

the authors' effort in the rebuttal was appreciated and led the reviewers to increase their scores.

For the camera-ready version, I suggest the authors improve the related work's sections, as many important citations are missing, particularly on the diffeomorphisms-based approach to stability and stability on the Riemannian manifolds (only citing the Euclideanizing Flows is not enough, as many other works in the area exist). Furthermore, it would be nice to include an in-depth discussion on the scalability of the method, particularly w.r.t. the diffeomorphism-based approaches.